# YouTube as a source of information on ectopic pregnancy: A qualitative and quantitative analysis

Camille Bulle [1] *, Marine Lallemant[1,2], Clara Rigori[1], Rajeev Ramanah[1,3]

1 Department of Obstetrics and Gynecology, University Medical Center of Besancon, University of Franche-Comte, Besançon, France, 2 Department of Applied Mechanics, FEMTO-ST Institute, UMR 6174 CNRS, University of Franche-Comte, Besançon, France, 3 Nanomedecine Laboratory, INSERM EA4662, University of Franche-Comte, Besancon, France

* bullecam@gmail.com

**Data Availability Statement:** All relevant data are within the paper and its Supporting Information files.

**Funding:** The author(s) received no specific funding for this work.

## Abstract

### Objective

The aim of this study was to assess the quality and reliability of YouTube videos on ectopic pregnancies for the public.

### Method

We searched for the key terms "ectopic pregnancy", "ectopic birth" and "extra uterine pregnancy" on YouTube. Each video that met the inclusion criteria was analyzed by two independent raters. Quantitative and qualitative metrics were recorded, and the videos were scored using the DISCERN instrument.

### Results

A total of 37 videos met the inclusion criteria. The mean overall DISCERN score was 44.5 ± 15.6. Videos had significantly a higher DISCERN score if they contained explanations on anatomy (pvalue <0.01), physiopathology (p-value <0.01), diagnosis (p-value = <0.01), treatments (p-value <0.01), symptoms (p-value <0.01), clear information (p-value <0.01), animations (p-value <0.01) and if it was a physician speaking (p-value <0.01).

### Conclusion

YouTube videos on ectopic pregnancy have been assessed to be only fairly reliable. We identified the five best ones using the validated DISCERN instrument. While ectopic pregnancy is not uncommon, YouTube videos should be improved to provide more accurate information for the public.

**Competing interests:** The authors have declared that no competing interests exist.

## Introduction

Nowadays, thanks to social networks and scientific popularization, 80% of the population has access to health information on the internet [1]. YouTube is the second most visited website in the world after Google [2] and the most popular video-sharing platform [3]. Besides normal pregnancy, the most common cause of pelvic pain and metrorrhagia with positive HCG hormone is ectopic pregnancy. It accounts for 2% of pregnancies. For the record, an ectopic pregnancy is defined as a pregnancy implanted outside the uterine cavity. It can be located in the fallopian tube (in 94% of cases and mainly in the ampullary part); but also in the cervix, the ovaries, the isthmus and even rarely intra-abdominally [4].

Ectopic pregnancy can be fatal if undetected, especially in developing countries [5]. According to the study by Laura L. Marion et al., the ectopic pregnancy mortality is around 3.9 of 1 000 which is 9% of all maternal deaths in USA [6]. In African developing countries, mortality raises between 1 and 3% of cases, making it one of the most important causes of maternal mortality [7].

Thus, about 75 to 80% of people use the internet for reassurance [8] because it is an infinite source of information. However, the question arises as to whether it can be trusted.

For all these reasons, we wondered about the quality of YouTube videos on ectopic pregnancies and what criteria influenced this quality.

The aim of this study was to assess the quality and reliability of YouTube videos on ectopic pregnancy for public. We also determined what information about ectopic pregnancies appealed to the audience, so that future educational video creators could better meet the needs of viewers and create more friendly videos. Finally, we tried to determine quality criteria for a public video.

## Methods

### Study design and data collection

The YouTube evaluation was conducted from 3 February 2022 to 14 April 2022. For this study, it was necessary to abide as closely as possible to the population. Therefore, we used non-technical keywords. We chose the following keywords: "ectopic pregnancy", "ectopic birth" and "extra uterine pregnancy".

The videos were ordered according to the default "relevance" sorting of YouTube. Only the first 30 videos for each keyword were evaluated, as some studies have shown that 90% of YouTube users looked at the results from the first three pages [9].

Only English-speaking videos were included because this language is considered universal [10]. All duplicate videos, unrelated to ectopic pregnancy, surgical videos or ultrasound videos without explanations were excluded.

### Variable extraction

The number of views, duration, number of comments, number of likes, source of the video were extracted. Other quantitative variables were obtained by vidIQ Vision for YouTube version 3.69.2 (Google Chrome extension) such as upload date, count of word and links in the description, referrers and tags, the number of views per hour, number of followers, number of followers per day and number of views of the channel. With the extension vidIQ Vision, we also extracted the VidIQ SEO score, which estimates the probability of the video appearing at the top of the search.

We calculated some information such as the average number of daily views with (total number of views / day since upload) and the likeability with two variables: the like ratio calculated by [like/ (like + dislike)] x100 and the video power index (VPI) calculated by [(like*100/

(like+dislike)) *(view/day)/100]. Since November 2021, dislikes are hidden to reduce cyberbullying. Thanks to a google extension named Return YouTube Dislike 2.1.0.3, we could extract this variable. We reported all data on Word Excel ®.

## Useful content

To determine whether a video is attractive and useful, we extracted some data common to other studies [11–13] such as data on anatomy, treatment, symptoms, physiology, whether there were animations on the slide video, how was made the diagnosis, how clear the information was and whether it was a physician speaking.

## Reliability and quality assessment

Videos were independently evaluated by two residents in Obstetrics and Gynecology who both had six years of experience in gynecologic emergencies. They were blinded to each other.

## DISCERN score

To report the reliability of videos, a validated instrument was used: the DISCERN scoring system [14, 15] (http://www.discern.org.uk/discern_instrument.php).

This score has been developed to help customers for evaluate the information health. The evaluation is mostly based on the presence of cited source.

It consists of 16 questions (Table 1); each question is scored as a scale from 1 to 5 with 1 indicating that the criteria are not met and 5 indicating that the criteria are fulfilled. The total of the 16 questions can be interpreted with the following scale: if the score is between 63 to 75, the reliability of the video is considerate excellent, between 51 to 62 as good, between 39 to 50 as fair, between 27 to 38 as poor, and between 26 to 16 as very poor.

## Statistics analysis

The mean and standard deviation were used as descriptive statistics for continuous variables. To compare two series of a quantitative variable, the Wilcoxon-Mann-Whitney test was performed. Intraclass correlation coefficient was carried out to ensure inter-rater reliability [16].

**Table 1. The DISCERN score.**

| |
|---|
| Q1: Are the aims clear? |
| Q2: Does it achieve its aims? |
| Q3: Is it relevant? |
| Q4: Is it clear what sources of information were used to compile the publication (other the author or producer)? |
| Q5: Is it clear when the information used or reported in the publication was produced? |
| Q6: Is it balanced and unbiased? |
| Q7: Does it provide details of additional sources of support and information |
| Q8: Does it refer to areas of uncertainty |
| Q9: Does it describe how each treatment works |
| Q10: Does it describe the benefits of each treatment? |
| Q11: Does it describe the risks of each treatment? |
| Q12: Does it describe what would happen if no treatment is used? |
| Q13: Does it describe how the treatment choices affect overall quality of life? |
| Q14: Is it clear that there may be more than one possible treatment choice? |
| Q15: Does it provide support for shared decision-making? |
| Q16: Based on the answers to all of the above questions, rate the overall quality of the publication as a source of information about treatment choices |

To assess the relationship between two quantitative variables, the Pearson correlation coefficient or Spearman correlation coefficient (rho) was calculated depending on whether the variable had a normal distribution or not.

R® version 4.1 software was used for the statistical analysis. For all analyses, we considered a p value less than 0.05 to be statistically significant.

### Ethical statement

No ethnical approval was necessary for this study because YouTube videos are publicly available.

The collection and analysis method complied with the terms and conditions for the source of the data.

## Result

### Videos characteristics

Between February 2022 and April 2022, 90 videos were identified but only 37 (41%) were included: 17 videos were duplicated, 2 were case reports, 3 were non-English speaking, 19 were off-topic, 7 were surgical videos, 4 were ultrasound courses, and one had no sound (Fig 1).

The most frequent source was medical channels (49%) which were mostly composed of gynecologist (27% of all videos). The second most frequent source was educative channels (24%) (Fig 2).

The average video length was 17 ±18 minutes. The mean likes and dislikes were 917 ± 2057 and 50 ± 133 respectively. On average, videos had 166.221 ± 482,310 views, 229.324 ± 445.247 channel subscribers, 74 ± 115comments, 14 ± 9 tags and 3 ± 8 references. Video descriptions contained 4 ± 6 links and 146 ± 182 words in average. The means time since the upload on YouTube and the beginning of this study was 38 ± 28 months (Table 2). The videos likability, the VPI and the score VidIQ were 94 ± 15 and 130 ± 305 and 45 ± 10 respectively.

**Videos contents.** The videos provided the following qualitative content: 27 (75%) videos described anatomy, 26 (72%) detailed symptoms, 22 (61%) specified treatments, 25 (69%) explained how the diagnosis could be done and 22 (61%) explained the physiology. Information seemed clear in 20 (56%) videos. A physician was speaking in 18 (50%) videos. There were animations in 23 (64%) videos (Fig 3).

**Video quality evaluation.** The mean overall DISCERN score was 44 ± 16. Thus, the global videos reliability was fair. There was no significant difference between the two raters (45 vs 44 p = 0.7) (Fig 4). The intraclass correlation coefficient (ICC) [16] of the DISCERN score was 0.81 (95% IC [0.63–0.90]). The inter-rater reliability was good.

Videos were ranked according to their global DISCERN score: 6 (16%) videos between 63 and 75, 7(19%) videos between 51 and 62, 8 (22%) videos between 60 and 39, 11 (30%) videos between 27 and 38, and 5 (13%) videos between 26 and 16.

The lowest average DISCERN score concerned the questions n˚ 4, 5 and 7 1.9 ±1.5, 1.8± 1.3, 1.3 ± 1.3 respectively (Fig 5). These three questions assessed whether or not the author claimed his sources. The highest average DISCERN score was given to the question n˚1: 4.4± 1.2. A high score at this question signified that video had a clear aim. Similarly, videos achieved their objectives (DISCERN score of question n˚2: 3.8 ± 1.5).

The question n˚6 had also a high DISCERN score (3.9 ± 1.6). It suggested that authors were impartial.

Video with a higher DISCERN score had significantly more explications about anatomy (p-value <0.01), about diagnosis (p-value <0.01), about treatment (p-value <0.01), about

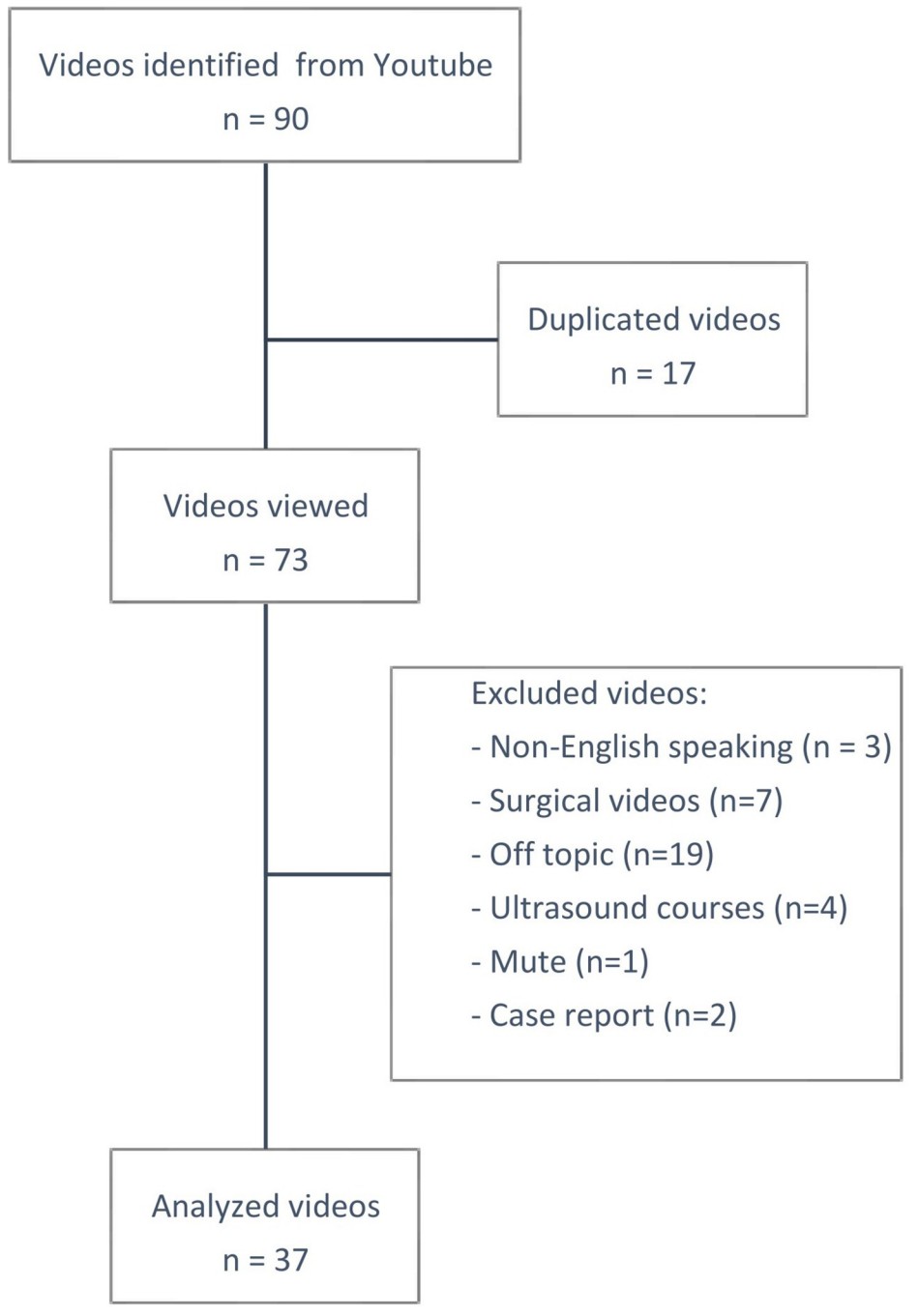

**Fig 1. Baseline features of the analyzed videos.**

symptoms (p-value <0.01), physiopathology (p-value <0.01) and had clear information (p-value <0.01).

Videos with a higher DISCERN score had significantly more animations (p-value <0.01) and presented by a physician (p-value <0.01).

There was no correlation between the DISCERN score and the vidIQ (p = 0.2), the video length (p = 0.1), the number of likes (p = 0.3) and dislikes (p = 0.4) and, the likability (p = 0.8).

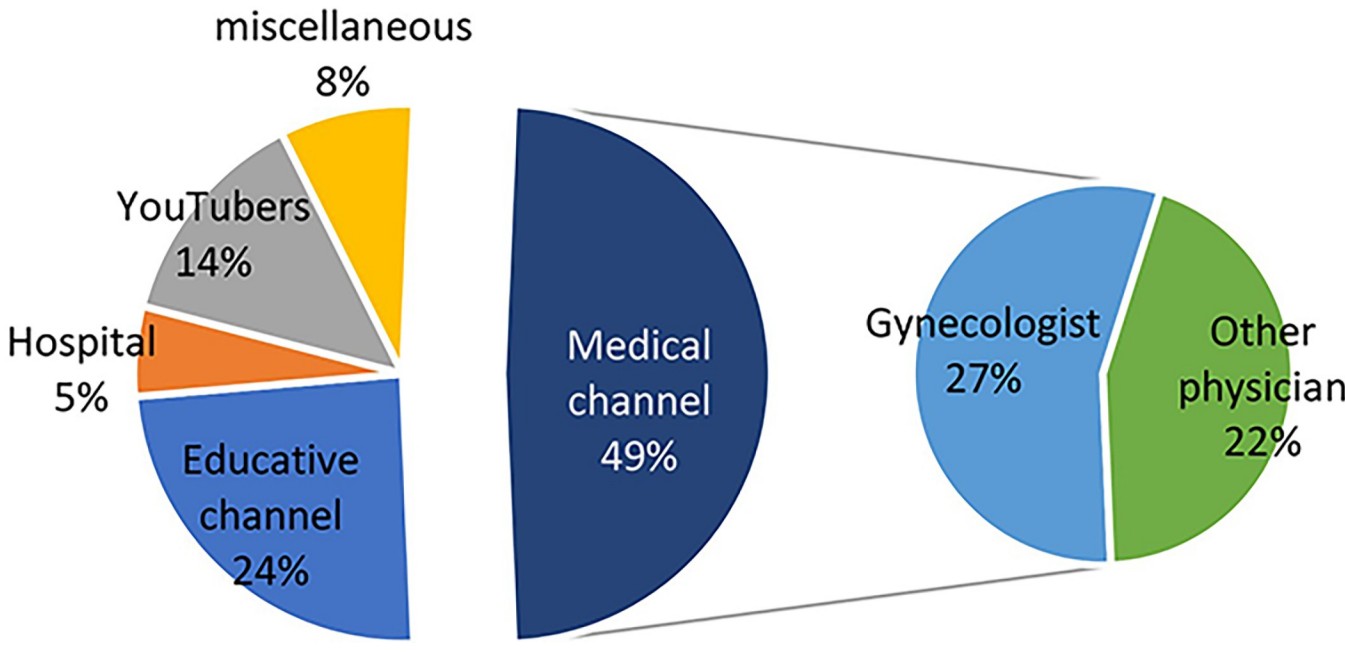

**Fig 2. Video sources.**

An optimization analysis was performed to determine the influence of animation, explanation about treatments, symptoms, anatomy, physiopathology, the presence of clear information, and a physician speaking, on the VPI, the video lengths, the likability, the number of comments and average daily view but none of the results were significative (p>0.05).

### The five best videos

Among these 37 videos, five videos had a DISCERN score greater than 63, which meant that their quality was excellent (Table 3). All of them were provided by a medical channel.

### Discussion

This study suggests that the quality of information on ectopic pregnancy was limited. With an average DISCERN score of 44, the reliability was fair. Sixteen (43%) videos had a poor or very poor DISCERN score (<38). The same results were found in similar studies assessing other diseases [11, 12, 17–19]. Only 37 of the 90 analyzed videos (41%) were included because the

**Table 2. Video characteristics.**

| Variables | Videos (n = 37) |
|---|---|
| Duration (minutes) | 17 ± 18 |
| Numbers of likes | 917 ± 25057 |
| Numbers of dislikes | 49.86 ± 132.7 |
| Numbers of views | 166 221 ± 482310.4 |
| Numbers of comments | 73.86 ± 9 |
| Numbers of months on YouTube | 38.35 ± 28,3 |
| Subscribers | 229 324 ± 445247.7 |

Data presented as mean and standard deviation

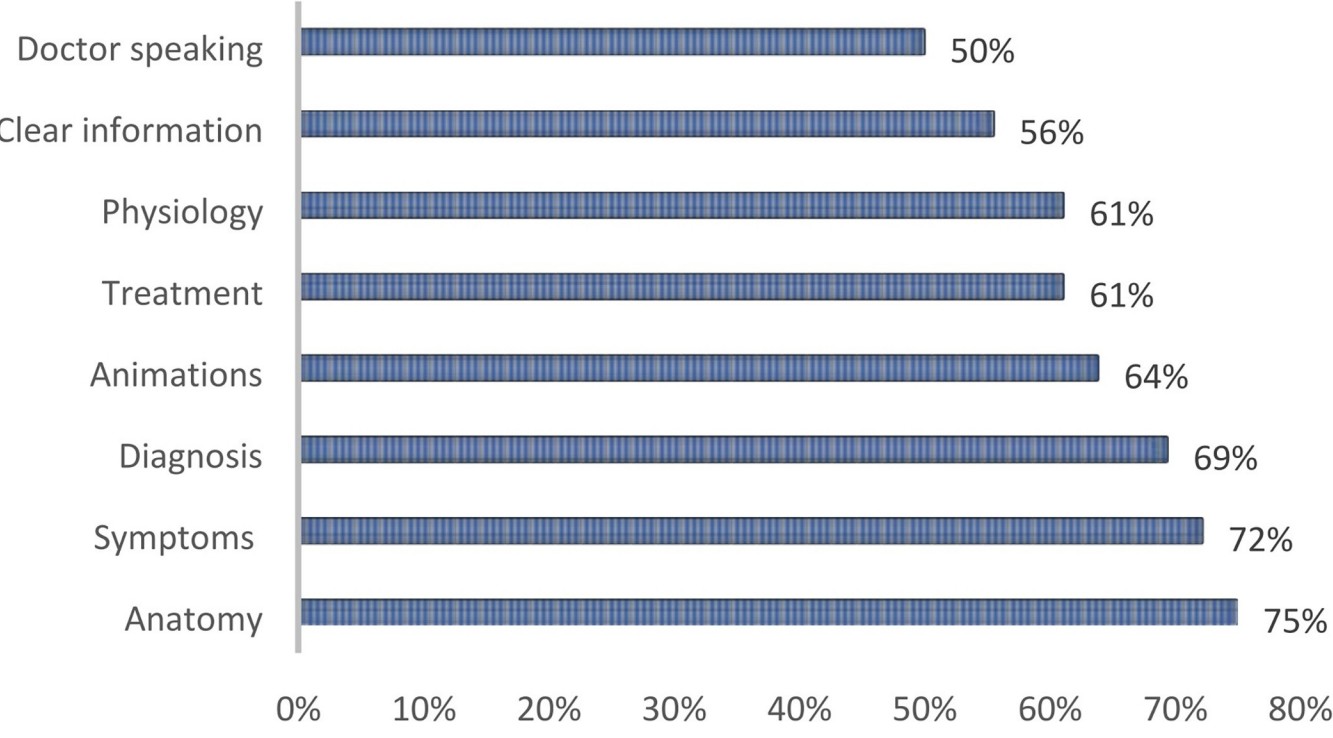

**Fig 3. Description of videos content.**

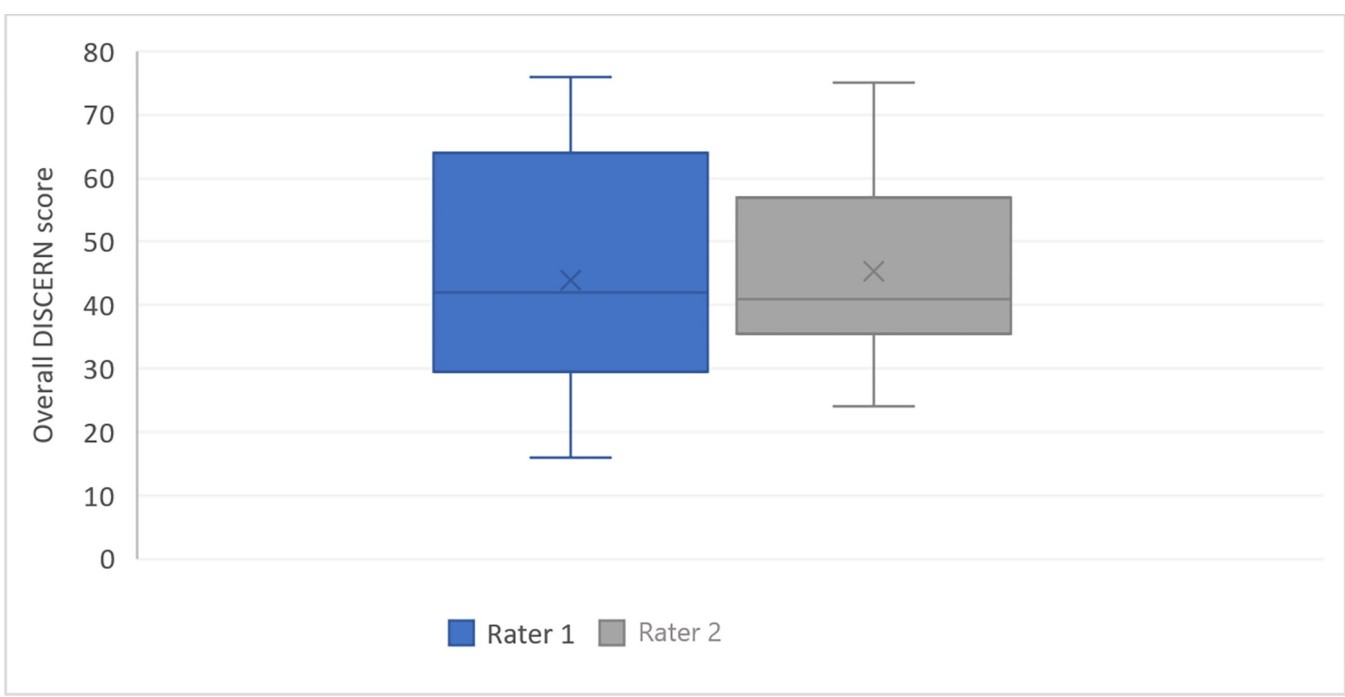

**Fig 4. Overall DISCERN score distribution among the two raters (p = 0.7).**

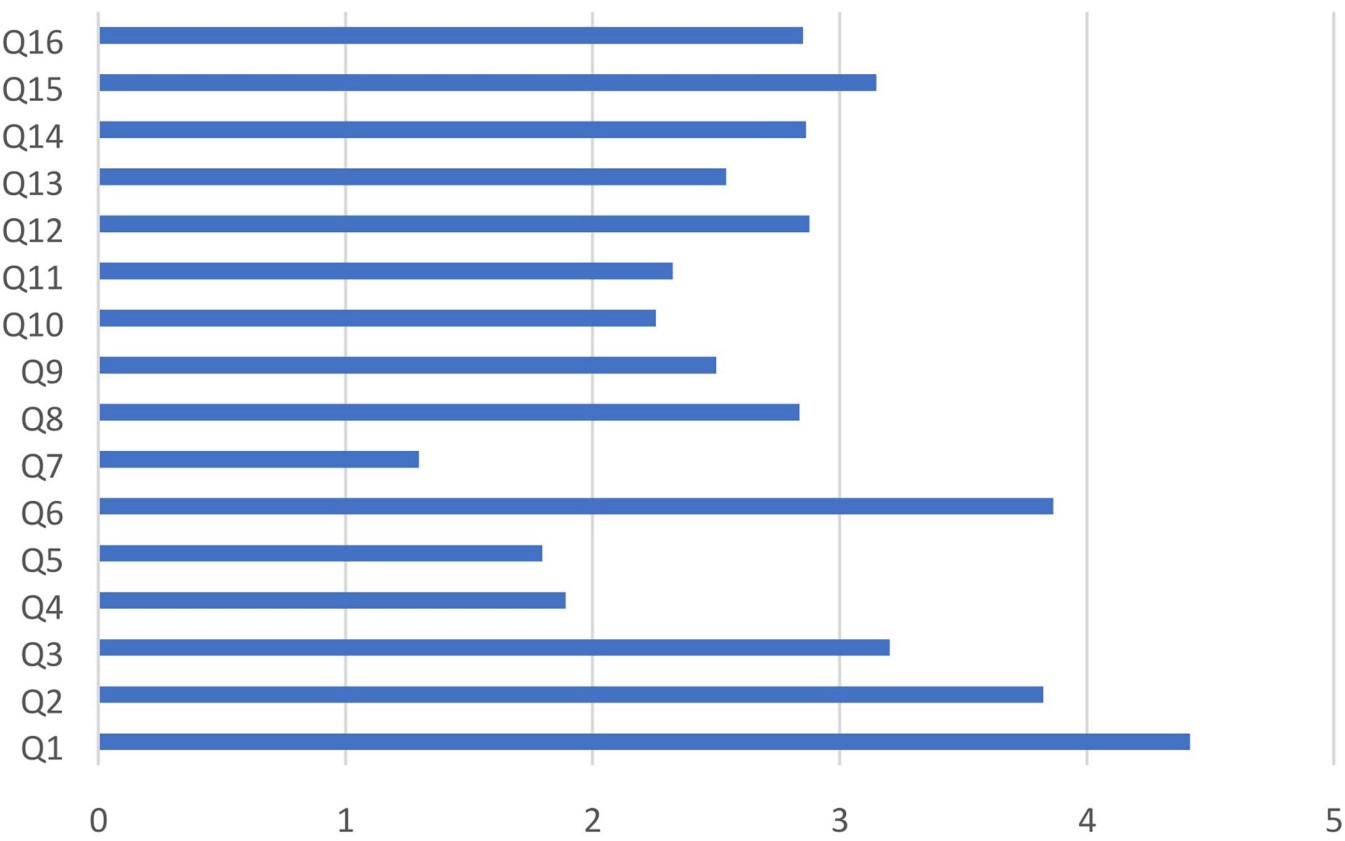

**Fig 5. Average DISCERN score for each question (Q = Question).**

information of the other videos was not usable. There were a lot of useless videos. Patient could get lost in them during their searches. Questions associated with low DISCERN scores (n° 4, 5 and 7) referred to the presence or not of explicit scientific sources. This result demonstrates that videos were rarely created from scientific data.

**Table 3. Five best videos.**

| DISCERN score | Title | Author | Year of production | Link |
|---|---|---|---|---|
| 71 | Learn OB/GYN: Ectopic Pregnancy | Learn OB/GYN | 2021 | ?v = QFXbWL-lfnQ&ab_channel = LearnOB%2FGYN |
| 69 | Ectopic Pregnancy | MedLecturesMa deEeasy | 2015 | watch?v=Q_qiUYIBR7o&ab_channel=MedLecturesMadeEasy |
| 65.5 | Ectopic pregnancy | Yasmin bano | 2021 | watch?v = MlanegSqHQo&ab_channel = yasminbano |
| 65 | Video Ectopic pregnancy | drsevellaraja | 2015 | watch?v = QUg0ZjDnUaM&ab_channel = drsevellaraja |
| 64.5 | Topic Ectopic Pregnancy | Association of Professor of Gynecology and Obstetrics (APGO) | 2015 | watch?v = AQBfRFmYQeA |

add https://www.youtube.com/ before the link to access to the video

Unsurprisingly, videos with high DISCERN score had more explanations about anatomy, diagnosis, treatment, symptoms, and had clearer information (p<0.01). Quality was higher when videos contained animations (p<0.01). According to the source, 49% of videos came from medical channels, and logically these videos containing a physician speaker had a higher score (p < 0.01). The same results were found in similar studies assessing other diseases [11, 12, 17–19]. These elements could be used to perform higher quality teaching videos in the future.

In contrast to the two studies published by Szmuda *et al.* evaluating the quality of YouTube information about narcolepsy and disk herniation, we did not find any argument to endorse YouTube's popularity [13, 20]. Indeed, Szmuda *et al.* demonstrated that videos obtained more comments and views if they contained explanations about anatomy and patient's experience. In our study, there was no significant association between the presence of information about anatomy, treatment, symptoms, physiology, whether there were animations on the slide video, how was made the diagnosis, how clear the information was and whether it was a physician speaking and, the video power index, the likeability, the duration, the number of comments and the average number of daily views.

Owing to this study, we were able to highlight the five best videos about ectopic pregnancy. Nevertheless, in these videos, the lexicon remains medical and is not always accessible to all audiences. This fact was also noticed by Szumda and *al.* [13].

We could note that the video with the highest DISCERN score was not the most popular. By the same way, the most popular video of our panel had a fair DISCERN score (48.5). This fact is also pointed out by Ward *et al.* [20]. The explanation would be that the authors of videos with high DISCERN scores would be physicians and contained more valid scientific information probably for scientific medical viewers. Thus, the most popular videos most were created by unqualified authors. They would be a source of misinformation. Our study supports the idea of Desai *et al.* [21]: people are not interested in scientific videos.

YouTube can be a great tool for health education because of its accessibility and popularity. It can greatly increase the reach of an educational program [22]. Improving the relationship between science and public, especially via the internet, is one of goals of the Global Sustainable Development Report (GSDR) 2023 [23].

Thanks to this study, we can draw up a list of recommendations for publishing an understandable video and allow awareness and knowledge sharing to the public. The video should be presented by a celebrity to reach the largest number of people. It should be done in collaboration with a health professional, a university or an association of professionals in order to quote reliable sources. Sources should appear distinctly to give the public the choice to obtain more information. In terms of content, videos should contain clearer explanations about anatomy, diagnosis, symptoms and treatment; all in an animated format.

This study was the first evaluating data from YouTube videos on ectopic pregnancy. One of the main strengths of this study was its high correlation coefficient (ICC = 0,81). Videos were analyzed by two experienced residents in Obstetrics and Gynecology. Moreover, this study was developed using the DISCERN instrument which has been validated by experts [14]. This tool is considered as a reproductive and objective tool.

This study had some limitations. First, we failed to establish a correct and appreciable length of videos. The literature suggested that the optimal length of educational videos would be less than 15 minutes [24].

Secondly, the method of analyzing the "clarity of information" data was subjective. The JAMA score could have been used to assess this element, but some studies have suggested that this score was also subjective [25, 26]. Lastly, YouTube's algorithm might skew the order of the top 30 videos, as the videos order changes following the viewing of previous videos. This

constitutes a selection bias. In fact, the order of the videos is random. This is the reason why we cannot say that there will not be new and more relevant videos using the algorithm. Ranade *et al.* [27] also noticed this bias. Like Madathil *et al.* [1], all these elements led us to believe that YouTube is not a reliable platform for medical information and could affect the clinician-patient relationship [28, 29].

## Conclusion

In this study, we showed that information from YouTube about ectopic pregnancy is limited. To improve its quality, videos should contain clearer explanations about anatomy, diagnosis, symptoms, and treatment. The video should be presented by a celebrity in partnership with a learned medical society. Sources should be cited. Studies have shown that the media and internet had a prominent place in the dissemination of information.

The popularization of medicine and healthcare through internet is useful but needs to be framed.

## Supporting information

**S1 Data.**
(XLSX)

**S2 Data.**
(XLSX)

## Author Contributions

**Conceptualization:** Camille Bulle, Marine Lallemant.

**Data curation:** Camille Bulle, Marine Lallemant, Clara Rigori.

**Investigation:** Camille Bulle, Clara Rigori.

**Supervision:** Marine Lallemant.

**Validation:** Rajeev Ramanah.

**Writing – original draft:** Camille Bulle.

**Writing – review & editing:** Marine Lallemant, Rajeev Ramanah.

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
