## [Decision Letter · Decision Letter 0]

15 May 2023

PONE-D-23-11462YouTube as a source of information  on ectopic pregnancy :  a qualitative and quantitative analysisPLOS ONE

Dear Dr. bulle,

Thank you for submitting your manuscript to PLOS ONE. After careful consideration, we feel that it has merit but does not fully meet PLOS ONE’s publication criteria as it currently stands. Therefore, we invite you to submit a revised version of the manuscript that addresses the points raised during the review process. Please consider all the comments made by the reviewers. 

Please submit your revised manuscript by Jun 29 2023 11:59PM. If you will need more time than this to complete your revisions, please reply to this message or contact the journal office at plosone@plos.org. Please include the following items when submitting your revised manuscript:A rebuttal letter that responds to each point raised by the academic editor and reviewer(s). You should upload this letter as a separate file labeled 'Response to Reviewers'.A marked-up copy of your manuscript that highlights changes made to the original version. You should upload this as a separate file labeled 'Revised Manuscript with Track Changes'.An unmarked version of your revised paper without tracked changes. You should upload this as a separate file labeled 'Manuscript'.

We look forward to receiving your revised manuscript.

Kind regards,

Angelica Espinosa Miranda, M.D., Ph.D.

Academic Editor

PLOS ONE

Journal Requirements:

2. In your Methods section, please include additional information about your dataset and ensure that you have included a statement specifying whether the collection and analysis method complied with the terms and conditions for the source of the data.

Reviewers' comments:

Reviewer's Responses to Questions

**Comments to the Author**

1. Is the manuscript technically sound, and do the data support the conclusions?

Reviewer #1: Partly

Reviewer #2: Yes

2. Has the statistical analysis been performed appropriately and rigorously? 

Reviewer #1: Yes

Reviewer #2: Yes

3. Have the authors made all data underlying the findings in their manuscript fully available?

Reviewer #1: No

Reviewer #2: Yes

4. Is the manuscript presented in an intelligible fashion and written in standard English?

Reviewer #1: Yes

Reviewer #2: Yes

5. Review Comments to the Author

Reviewer #1: Report of the article " YouTube as a source of information on ectopic pregnancy : a qualitative and quantitative analysis".

The article "YouTube as a source of information on ectopic pregnancy : a qualitative and quantitative analysis" deals with a contemporary topic of great interest, especially for those who work with non-formal health education.

Recommendation to authors:

[1. Introduction

The authors need to improve the introduction of the article a lot.

It is necessary to better contextualize the problem and present epidemiological data on ectopic pregnancy. It is not enough simply to state that this can be a fatal pregnancy if left undetected, and that this is mainly a problem in developing countries. I consider it fundamental that the authors present data on ectopic pregnancy in some countries, this will qualify the statements made. For example, show data on deaths and ectopic pregnancies in developed and developing countries.

In the introduction, the authors should make clear the objective of the article and present the research questions.

[2] Methodology

The methodology is well described, however, I recommend that the authors place in a supplementary file a list of all the videos that were selected for the study. This is important to ensure method replication as well as research transparency.

Authors should create a subsection to better explain the DISCERN tool.

[3] Related works

Authors must include a related work section. It is important to know if there are other works and what these works are doing different from the one presented by the authors.

The related work section should be a section where the authors will deepen the discussion on topics based on the works found.

Authors can use more recent works, between 2019 to 2023 and make a critical analysis of the works. The authors state in the discussion that this article was the first to evaluate data from YouTube videos on ectopic pregnancy. However, to better qualify this statement, the authors must present it in this section of related works.

[4] Discussions

The authors make the following statement in the discussion: "YouTube’s algorithm might skew the order of the top 30 videos, as the videos order changes following the viewing of previous videos. This constitutes a selection bias."

I agree that the display of videos changes depending on the YouTube algorithm. However, why would that be a problem? The authors should explain this part better - only the issue of bias in the selection of videos is not enough.

Finally, the authors must include in the discussion a list of recommendations so that a video has quality and at the same time can dialogue (be understandable) with the population (accessible content). It is necessary to deepen this point a little in the discussion and talk about non-formal education in health. The authors talk about this in the conclusions, but in a very superficial way.

Reviewer #2: 

Dear Authors

I begin by congratulating the authors for the work presented.

Great impact on women's health, mental health and public health in general.

Some questions:

- in table 2 the number of likes is one of the most complex variables. Very asymmetrical;

- in table 3, I suggest you include the year of production of the video (the content may be less or more updated);

Conclusions can be improved. as a suggestion the involvement of the university and health professionals in the execution of these videos.

They must include the sustainable development goals (Agenda 2023) to reinforce the pertinence of the theme and the objectives of this article.

6. PLOS authors have the option to publish the peer review history of their article (what does this mean?). If published, this will include your full peer review and any attached files.

Reviewer #1: **Yes: **Ricardo Alexsandro de Medeiros Valentim

Reviewer #2: **Yes: **Aliete Cunha-Oliveira

---

## [Author Response · Author response to Decision Letter 0]

23 May 2023

May 18, 2023, 

Object: My response to the comments from the reviewers 

Manuscript ID: PONE-D-23-11462R1

Title: YouTube as a source of information on ectopic pregnancy: a qualitative and quantitative analysis

Dear Angelica Espinosa Miranda,

We thank you and the reviewers for your time in reviewing our article. I carried out the corrections as suggested by reviewers. You will find below my response to the comments.

Reviewer 1. 

Introduction

The authors need to improve the introduction of the article a lot.

It is necessary to better contextualize the problem and present epidemiological data on ectopic pregnancy. It is not enough simply to state that this can be a fatal pregnancy if left undetected, and that this is mainly a problem in developing countries. I consider it fundamental that the authors present data on ectopic pregnancy in some countries, this will qualify the statements made. For example, show data on deaths and ectopic pregnancies in developed and developing countries.

All these suggestions have been considered and appear highlighted in the text (lines 82 – 85).

In the introduction, the authors should make clear the objective of the article and present the research questions.

All these suggestions have been considered and appear highlighted in the text (lines 94– 95 and 100)

Methodology

The methodology is well described, however, I recommend that the authors place in a supplementary file a list of all the videos that were selected for the study. This is important to ensure method replication as well as research transparency.

The supplementary file has been uploaded as “YouTube videos link”.

Authors should create a subsection to better explain the DISCERN tool.

A subsection has been created. (line 155).

 Related works 

Authors must include a related work section. It is important to know if there are other works and what these works are doing different from the one presented by the authors.

The related work section should be a section where the authors will deepen the discussion on topics based on the works found.

Authors can use more recent works, between 2019 to 2023 and make a critical analysis of the works. The authors state in the discussion that this article was the first to evaluate data from YouTube videos on ectopic pregnancy. However, to better qualify this statement, the authors must present it in this section of related works.

 After a long review of the literature, there are very few studies about YouTube information, and none about YouTube videos and ectopic pregnancy. The methodology applied in this article is the same as in the other articles dealing with the quality of YouTube videos on other topics. The only differences are explained in the discussion section.

 Discussions

The authors make the following statement in the discussion: "YouTube’s algorithm might skew the order of the top 30 videos, as the videos order changes following the viewing of previous videos. This constitutes a selection bias."

I agree that the display of videos changes depending on the YouTube algorithm. However, why would that be a problem? The authors should explain this part better - only the issue of bias in the selection of videos is not enough.

 The suggestion has been considered. ( line 342)

Finally, the authors must include in the discussion a list of recommendations so that a video has quality and at the same time can dialogue (be understandable) with the population (accessible content). It is necessary to deepen this point a little in the discussion and talk about non-formal education in health. The authors talk about this in the conclusions, but in a very superficial way.

The suggestions have been considered. (line 328)

Reviewer 2

Dear Authors

I begin by congratulating the authors for the work presented.

Great impact on women's health, mental health and public health in general.

Some questions:

- in table 2 the number of likes is one of the most complex variables. Very asymmetrical;

The table has been modified. 

- in table 3, I suggest you include the year of production of the video (the content may be less or more updated);

The table has been modified. 

Conclusions can be improved. As a suggestion the involvement of the university and health professionals in the execution of these videos.

They must include the sustainable development goals (Agenda 2023) to reinforce the pertinence of the theme and the objectives of this article.

The suggestions have been considered. (line 317)

Now that the corrections are made, I hope you will find the manuscript acceptable for publication in PLOS ONE Journal. Please feel free to contact us with any questions or concerns.

Sincerely,

Camille BULLE

---

## [Decision Letter · Decision Letter 1]

8 Jun 2023

YouTube as a source of information  on ectopic pregnancy :  a qualitative and quantitative analysis

PONE-D-23-11462R1

Dear Dr. Camille Bulle,

We’re pleased to inform you that your manuscript has been judged scientifically suitable for publication and will be formally accepted for publication once it meets all outstanding technical requirements.

Kind regards,

Angelica Espinosa Miranda, M.D., Ph.D.

Academic Editor

PLOS ONE

Additional Editor Comments (optional):

Reviewers' comments:

Reviewer's Responses to Questions

**Comments to the Author**

1. If the authors have adequately addressed your comments raised in a previous round of review and you feel that this manuscript is now acceptable for publication, you may indicate that here to bypass the “Comments to the Author” section, enter your conflict of interest statement in the “Confidential to Editor” section, and submit your "Accept" recommendation.

Reviewer #1: All comments have been addressed

Reviewer #2: All comments have been addressed

2. Is the manuscript technically sound, and do the data support the conclusions?

Reviewer #1: Yes

Reviewer #2: Yes

3. Has the statistical analysis been performed appropriately and rigorously? 

Reviewer #1: Yes

Reviewer #2: Yes

4. Have the authors made all data underlying the findings in their manuscript fully available?

Reviewer #1: Yes

Reviewer #2: Yes

5. Is the manuscript presented in an intelligible fashion and written in standard English?

Reviewer #1: Yes

Reviewer #2: Yes

6. Review Comments to the Author

Reviewer #1: The authors endeavored to meet the recommendations. Therefore, I am in favor of accepting the article for publication in this journal.

[Only as a suggestion] I believe that the authors could still improve the conclusions for the last version of the article, before publication. For example, talking about future work on this topic, what would be the new paths for research in this area and what is the importance of this model of non-formal education.

Reviewer #2: Congratulations to the authors.

They effectively answered the questions asked. Making it a complete article.

Article relevant to science.

7. PLOS authors have the option to publish the peer review history of their article (what does this mean?). If published, this will include your full peer review and any attached files.

Reviewer #1: **Yes: **Ricardo Valentim

Reviewer #2: **Yes: **Aliete Cristian Gomes Dias Pedrosa da Cunha-Oliveira

---

## [Editor Report · Acceptance letter]

4 Jul 2023

PONE-D-23-11462R1 

YouTube as a source of information  on ectopic pregnancy :  a qualitative and quantitative analysis 

Dear Dr. Bulle:

I'm pleased to inform you that your manuscript has been deemed suitable for publication in PLOS ONE. Congratulations! Your manuscript is now with our production department. 

Kind regards, 

on behalf of

Dr. Angelica Espinosa Miranda 

Academic Editor

PLOS ONE